# Garlic Alleviates the Injurious Impact of Cyclosporine-A in Male Rats through Modulation of Fibrogenic and Steroidogenic Genes

**DOI:** 10.3390/ani11010064

**Published:** 2020-12-31

**Authors:** Mustafa Shukry, Saqer S. Alotaibi, Sarah M. Albogami, Nora Fathallah, Foad Farrag, Mahmoud A. O. Dawood, Mahmoud S. Gewaily

**Affiliations:** 1Department of Physiology, Faculty of Veterinary Medicine, Kafrelsheikh University, Kafrelsheikh 33516, Egypt; 2Department of Biotechnology, College of Science, Taif University, P.O. Box 11099, Taif 21944, Saudi Arabia; saqer@tu.edu.sa (S.S.A.); dr.sarah@tu.edu.sa (S.M.A.); 3Department of Zoology, Faculty of Science, Kafrelsheikh University, Kafrelsheikh 33516, Egypt; nora_fathalla@yahoo.com; 4Department of Anatomy and Embryology, Faculty of Veterinary Medicine, Kafrelsheikh University, Kafrelsheikh 33516, Egypt; foad.farrag@yahoo.com (F.F.); drmahmoud_gewaily@yahoo.com (M.S.G.); 5Department of Animal Production, Faculty of Agriculture, Kafrelsheikh University, Kafrelsheikh 33516, Egypt; mahmoud.dawood@agr.kfs.edu.eg

**Keywords:** rat, cyclosporine-A, garlic, fibrosis, steroidogenesis

## Abstract

**Simple Summary:**

The use of garlic extract is a conventional approach in improving the side effects induced by cyclosporine A (CsA) and in maintaining health. The current study explored the impact of garlic on liver and testicular function, blood biochemical parameters, and oxidative stress in rats raised under cyclosporine A toxicity. It was found that the administration of garlic restored liver function and modulated lipid markers. Garlic supplementation improved the gene expression of Superoxide dismutase (SOD) and steroidogenesis genes, decreased that of collagen I-α1 (Col1a1) and transforming growth factor-1β (TGF-β1), and enhanced the antioxidant status and fertility. A combined treatment of garlic and CsA is advocated to alleviate CsA-induced oxidative stress injuries and other adverse effects.

**Abstract:**

This work aimed to study the hepato-testicular protective effect of garlic in rats treated with cyclosporine A (CsA). Forty male Westar albino rats were randomly distributed in five groups (8 rats each): control, olive oil, garlic, CsA, and CsA co-treated with garlic. CsA induced an upsurge in the alanine transaminase, aspartate transaminase, and alkaline phosphatase levels and decreased albumin and total protein levels, expression of superoxide dismutase (SOD) gene, serum testosterone, triiodothyronine, and thyroxine levels compared to the control group. Additionally, there was an increase in the cholesterol, triglyceride, and low-density lipoprotein levels and a substantial reduction in the high-density lipoprotein levels compared to the control groups. Histopathological investigation of the liver showed abnormalities like hepatic cell degeneration, congestion of blood vessels, and highly active Kupffer cells in the CsA group. Histopathological examination of testes showed damaged seminiferous tubules, stoppage of the maturation of spermatogonia, and the presence of cells with irregular dense nuclei in the lumina of some tubules. For the groups treated with garlic, mitigation of the damage caused by CsA in the liver and testes, liver function tests, lipid profiles, and hormones was seen along with improved gene expression of SOD and steroidogenesis genes, and decreased gene expression of collagen I-α1 and transforming growth factor-1β. Conclusively, garlic had a positive impact on CsA-induced hepatic and sperm toxicity. It is recommended that garlic should be supplemented in transplant treatments using CsA to alleviate the cyclosporin-induced oxidative injuries and other harmful effects.

## 1. Introduction

The liver plays a crucial role in controlling body homeostasis. It is involved in almost all the biochemical development, nutrient supply, energy production, and disease resistance pathways. It is also an essential target for drug intoxication, xenobiotics, and oxidative stressors [1]. Liver cells are interconnected by releasing common mediators like growth and reactive oxygen species (ROS), which stimulate liver fibrosis and release TGF-β1 and collagen I-α [2,3]. Damage to the liver caused by hepatotoxic agents may therefore have severe implications. Most hepatotoxic compounds induce lipid peroxidation and other oxidative disruptions [4].

Cyclosporine A is a fungal peptide. It is most widely used as a strong immunosuppressant in transplant operations and autoimmune diseases due to its unique inhibitory impact on the T cell’s lymphokine production and signal transduction. The clinical use of CsA is, however, restricted to side effects [5]. Previous studies have shown that CsA has caused ovarian damage [6], testicular [7,8], spermatozoal damage, and spermatozoal toxicity [9,10,11,12,13,14]. The administration of CsA has been reported to cause a dose-dependent decrease (20 mg/kg or higher) in the reproductive organ weights male rats [10,12,14], as well as a reduction in the level of testosterone [15,16], decreases in the diameter of germinal cell thickness., and the deceleration of spermatogenesis was also seen in the lumen of certain Seminiferous tubules T of CsA-treated rats [17]. Earlier findings suggest hypothalamic-pituitary-gonadal axis alterations [13,14,15] and reduction in Sertoli cell phagocytic function [8,17,18] caused by CsA administration is maybe accountable for the pathogenesis of testicular and spermatozoal toxicity.

Cyclosporine-A (CsA), in particular, causes severe side effects, like nephrotoxicity, hepatotoxicity, hypertension [19], and sperm toxicity [17], which often limits the use of the drug. The body has developed a highly advanced, complex antioxidant defense mechanism for cells and organs against ROS. This framework consists of various elements, particularly antioxidants that collaborate and simultaneously counteract free radicals [20]. Superoxide dismutases (SODs) are a class of enzymes that catalyze the degradation of the superoxide anion to oxygen and hydrogen peroxide [21]. These enzymes are used in almost all aerobic cells and extracellular fluids [22]. SOD enzymes have metal ion cofactors like copper, zinc, manganese, or iron, depending on the isozyme [23]. Herbs have been shown to play a substantial role in treating different liver [24] and reproductive toxicity in testes [25]

Garlic (*Allium sativum* L.) is one such herb that has gained a standing as a preventive and curative medicinal plant in Sumerian, ancient Egyptian, Chinese, and Indian medicinal traditions [26]. As a medicinal plant, garlic enhances immune function, and has antiviral, antimicrobial, antioxidant, and anti-inflammatory properties. The antioxidative actions of garlic and its components, such as allixin, S-allylcysteine (SAC), S-allylmercaptocysteine (SMAC), and diallyl polysulfides, are determined by their ability to scavenge ROS, and this ability increases with the number of sulfur atoms. Some of these antioxidative actions are the inhibition of the formation of lipid peroxides [27,28], and diminishing the risk of heart disease and cancer [29] by lowering blood pressure and cholesterol levels. Garlic and its components are also used to treat the common cold [30]. In terms of its efficacy in preventing adverse side effects related to liver function, garlic contains organosulfur and diallyl disulfide compounds, which are liver-protective compounds [31]. These compounds also repress cytochromes P450 (CYP17A1), which are responsible for the biological activation and generation of deleterious oxyradicals for a wide range of hepatotoxins [32]. Moreover, organosulphur compounds enhance activities that are important for the hepatic detoxification processes of phase II enzymes, such as S-transferases glutathione, UDP-glucuronyl transferase, and microsomal epoxide hydrolase [33] as well as glutathione peroxidase and dismutase superoxide activities, which are enhanced by the organosulfur compounds present in garlic oil [34]. The goal of the current study was, therefore, to study the protective effects of garlic against oxidative injuries induced by CsA, biochemical alterations, and DNA impairment.

## 2. Materials and Methods

### 2.1. Animals and Ethical Statement

The experimental procedures were approved by the Institutional Animal Care and Use Committee at Kafrelsheikh University, Egypt (Number-2018-/1155). All precautions were taken during the process to alleviate animal suffering. Forty male Wistar albino rats (average weight 130 ± 15 g) that were approximately one and a half months of age were used during this experiment. The animals were housed in wire mesh cages, fed on a daily diet of rodent pellets, and had free access to tap water. Rats were held in the laboratory for 15 days for adaptation at 23 ± 2 °C with a relative humidity of 55 ± 5% and a 12/12 h light/dark cycle.

### 2.2. Chemicals

Cyclosporine-A (Sandimun Neoral^®^ capsules were purchased from Novartis Pharma Co, Plantation, FL, USA) was dissolved in 2 mL of olive oil. Garlic was administered to the mice through Tomex tablets that were purchased from Atos Pharma Co., Cairo, Egypt.

### 2.3. Experimental Design

The experimental procedures were performed at the Physiology Department, Faculty of Veterinary Medicine, Kafrelsheikh University. Following adaptation, the rats were alternately and equally allocated into five equal groups (8 rats each). All chemicals were administered orally using a stomach tube. Group 1 (Control group; CTR) rats received saline (2 mL/kg per day) an equivalent amount of saline according to each treated rat’s weight for four weeks. Group 2 (Placebo group; PAL) rats received olive oil (2 mL/kg per day), an equivalent amount of olive oil according to each treated rat’s weight for four weeks. Group 3 (Garlic group; GAR) was the positive control group in which the rats were administered garlic (40 mg/kg per day; dissolved in saline) for four weeks. The Garlic dose was chosen according to the previous studies [35,36]. Garlic extract: 200 mg of garlic tablet were weighted by sensitive balance then homogenized with 10 mL saline then take 2mL of this mixture to reach the garlic’s desired dose (40 mg/kg) then administered by oral gavage to each rat according to each weight. Group 4 (Cyclosporine-A group; CsA) rats were administered CsA (10 mg/kg per day) according to [37,38]. The cyclosporine capsule (25 mg) dissolved in 5 mL olive oil, then take 2mL of this mixture to reach the desired dose of the cyclosporine (10 mg/kg), then administered by oral gavage to each rat of the Cyclosporine group according to each weight for four weeks. Group 5 (a co-treated group with garlic and cyclosporine-A; GAR + CsA) rats were administered CsA (10 mg/kg per day) and garlic (40 mg/kg per day) for four weeks (see Figure 1, experimental design).

### 2.4. Collection of Samples

After overnight fasting at the end of the experiment, the rats were anesthetized by CO_2_ inhalation and euthanized by exsanguination. Blood samples were collected after severing the jugular veins. The serum samples were aspirated and frozen until analysis. The liver and testis were quickly removed, washed with cold saline to remove extraneous material, and then the specimens were fixed in formalin solution (10%) for the first day, and then moved to a solution of 70% alcohol. Tissues of the liver and testes were obtained from all groups. The tissue samples were placed in 2 mL Eppendorf tubes and immediately stored at −80 °C before extraction of RNA.

### 2.5. Biochemical Analysis

The alanine transaminase (ALT) (Cat. No. AL1031 (45)), and aspartate transaminase (AST) (Cat. No. AS1061(45)) activities were calorimetrically determined using commercial kits, according to the protocol followed by Reitman and Frankel [39]. The alkaline phosphatase (ALP) (Cat. No. AP1020) activity was determined according to the protocol followed by Kind and King [40]. The total protein level (Cat. No. TP2020) and albumin level (Cat. No. AB1010) were assayed using commercially supplied kits, according to Bowers and Wong [41]. Serum total cholesterol (Cat. No. CH1220) and serum triglycerides (Cat. No. TR2030) were determined according to methods described previously [42,43]. The levels of serum high-density lipoprotein cholesterol (HDL-c) (Cat. No. CH1230) were estimated according to a method described previously [44]. However, the levels of low-density lipoprotein cholesterol (LDL-c) (Cat. No. CH1231) were assessed according to a protocol described by Friedewald et al. [45]. All the kits were commercially supplied by Biodiagnostic (Cairo, Egypt). Serum testosterone was determined according to Dunn et al. [46] using ARCHITECT Testosterone Reagent Kit (7K73), and serum T3 and T4 were determined according to Abuid et al. [47] using the ARCHITECT Total T3 Reagent Kit (7K64) and ARCHITECT Total T4 Reagent Kit (7K66), respectively.

### 2.6. Sperm Count, Motility, Viability, and Morphology

The sperm count was measured microscopically using a hemocytometer according to the protocol followed by Yokoi et al. [48] after mincing the cauda epididymis in 5-mL saline and diluting the supernatant in an alkaline aqueous solution. Sperm motility and live sperm were evaluated as previously described [49]. Viability testing was performed using a dye exclusion method. For sperm abnormality evaluation, each epididymis content was mixed with an eosin-nigrosin stain drop and a thin film was smeared over clean slides, with a random examination of 300 spermatozoa per slide [50].

### 2.7. Molecular Investigation

The RNA was extracted from liver tissues; pure RNA was obtained using the RNA Purification Kit (Thermo Scientific, Fermentas, #K0731). Complementary DNA (cDNA) was obtained using Revert Aid H minus Reverse Transcriptase, a genetically adjusted M-MuLV RT that converted the RNA into complementary DNA (cDNA). SYBR Green–real-time PCR assay was used to assess the expression of the mRNAs of the liver and testes genes with internal references to β-actin (Table 1).

### 2.8. Histopathological Examination

After fixation, the specimens were dehydrated by passing them through a series of alcohol solutions in ascending concentrations up to 100%, after which the alcohol was cleared in two xylene shifts, and the specimen was enclosed in molten paraffin. 5-micron thick sections were cut with a rotary microtome and attached to clean slides. The sections were stained using Ehrlich’s hematoxylin and eosin [51].

### 2.9. Data Analysis

Data were obtained, measured, analyzed, and expressed as mean ± standard error of mean (SEM) using one-way analysis of variance (ANOVA) (SPSS, 18.0 software). For individual comparisons, Duncan’s multiple-range test (DMRT) was used. *p* < 0.05 was considered statistically significant for all the data.

## 3. Results

### 3.1. Behavioral Pattern

In animals treated with CsA, various side effects were observed, like decreased feed and water consumption, loss of body weight and activity, weakness, yellowish eyes and gums, loss of body hair, and bleeding from the nose and eyes. The recorded mortality rate of the CsA group (group 4) was 37.5% (3 out of 8 rats); no mortality was recorded among the other groups.

### 3.2. Biochemical Investigations

As shown in Table 2, rats that received olive oil (PAL) and garlic (GAR) had non-substantial alterations in the ALT, AST, and ALP levels compared to rats from the control group. In contrast, the rats in the CsA group showed significant increases in these levels compared to the control group. Conversely, the rats co-treated with garlic (GAR + CsA) showed a considerable decrease (*p* < 0.05) in these levels compared to the CsA-treated rats, although their levels were not identical to that of the control rats.

Total protein and albumin serum concentrations declined substantially (*p* < 0.05) in the CsA group compared to the control group. However, rats co-treated with garlic and CsA (GAR + CsA) exhibited substantial upregulation (*p* < 0.05) in these levels compared to the CsA group. Additionally, the serum levels of albumin and total protein in the PAL and GAR groups indicated a non-significant difference compared to the control group.

Table 3 indicates that there was no substantial difference in the total cholesterol, triglycerides, and LDL-c serum levels in the PAL and GAR groups compared to the control group. Nevertheless, these levels increased considerably (*p* < 0.05) in the CsA group compared to the control group. Additionally, the GAR + CsA group showed a considerable reduction (*p* < 0.05) in these levels compared to the CsA-treated group. Moreover, the HDL-c levels were reduced significantly (*p* < 0.05) in the CsA-treated group compared to the control group, although these levels exhibited substantial improvement (*p* < 0.05) in the GAR + CsA group. Furthermore, serum levels of HDL-c in the PAL and GAR groups showed non-substantial differences compared to the control group.

### 3.3. Reproductive Hormones and Seminal Analysis

Table 4 shows that the serum levels of testosterone, triiodothyronine (T3), and thyroxine (T4) exhibited non-substantial variations in the PAL and GAR groups compared to the control group. On the other hand, rats in the CsA group showed a considerable reduction (*p* < 0.05) in these levels compared to their control group counterparts. Notably, rats in the GAR + CsA group showed significant upregulation (*p* < 0.05) in these hormone levels, although these were not identical to the rats in the control group. Table 5 shows a significant increase in sperm cell concentration, sperm cell motility, and sperm cell viability in the PAL and GAR groups compared to the CsA (*p* < 0.05) group, with a significant increase in the sperm cell abnormalities in the CsA (*p* < 0.05) group compared to the other treated groups.

### 3.4. mRNA Gene Expression

Figure 2A shows that the mRNA expression of the SOD_1_ gene was significantly downregulated in the group treated with CsA compared to the control group. However, co-treatment with garlic resulted in a substantial increase in the liver tissue gene expression compared to the rats treated solely with CsA. Rats supplemented with olive oil (PAL) and garlic (GAR) did not have any significant variations in the gene expression compared to the CsA group, which displayed substantial upregulation in the mRNA expression of both collagen I-α and TGF-β1 compared to the control group. Meanwhile, co-treatment with garlic and CsA noticeably decreased the mRNA expression of collagen I-α and TGF-β1. Garlic administration alone resulted in a significant decline in both the fibrosis-related genes. However, thioredoxin mRNA (Txn-1) expression was significantly decreased in the CsA group, but significantly improved in the GAR+CsA group. Concerning steroidogenesis genes, as shown in Figure 2B, the co-administration of garlic and CsA substantially improved the mRNA expression of the STAR gene along with restoring the normal level of the CYP17A and 3β-HSD genes comparing with the control group, which was expected as the garlic group (GAR) alone showed a substantial increase in the mRNA levels that were greatly reduced in the CsA group.

### 3.5. Histopathologic Studies

Microscopic examination of the liver tissues in the control, placebo, and garlic-treated groups revealed typical architecture in the hepatic lobules, central veins, and portal triads (Figure 3A,B). Meanwhile, liver tissues from rats treated with CsA showed multifocal hemorrhage areas, degeneration, and necrosis, together with congested blood vessels (Figure 3C). In contrast, rats in the GAR+CsA group showed nearly normal hepatic lobules, portal areas, and vasculature with mild activation of the Kupffer cells (Figure 3D). Microscopic examination of testicular tissues from the control, placebo, and garlic-treated rats revealed normal histologic limits of the seminiferous tubule and their lining, germinal epithelium, sperm aggregations within the tubular lumina, and interstitial tissue, including blood vessels and the Leydig cells (Figure 3E,F). However, examination of testicular tissues from rats treated with CsA revealed damage to the seminiferous tubules, arrested maturation up to the spermatogonia stage, and vast intertubular space (Figure 3G). In the GAR + CsA group, testicular tissue examination revealed a moderate improvement of the seminiferous tubules, with some showing complete maturation and narrow interstitial spaces with normal blood vessels (Figure 3H).

## 4. Discussion

CsA is a potent immunosuppressant widely used to treat organ transplants and multiple autoimmune illnesses [52]. However, it is to be noted that patients tend to take this drug lifelong. Owing to its adverse effects, including hepatotoxicity, nephrotoxicity, cardiotoxicity, and sperm toxicity, the safety of extensive CsA use is debatable [53]. The liver is the primary organ responsible for drug absorption and detoxification of toxic chemicals. Thus, it is a critical organ [54,55], and efforts to discover an effective treatment for liver injuries are crucial in clinical scenarios. Consequently, natural medicines or compounds such as those present in garlic, can be utilized, which can antagonize the deleterious action of free radicals and protect hepatocytes from damage via antioxidant activities. Garlic is used as an antioxidant to prevent the generation of free radicals, support body-protective mechanisms that destroy free radicals [56], and avoid hypocholesterolemia [57]. The present study assessed the magnitude of cyclosporine toxicity on liver functions in rats by determination of the ALT, AST, and ALP levels (Table 1), in which rats supplemented with olive oil (PAL) and garlic (GAR) showed non-significant alterations in ALT, AST, and ALP levels compared to the control group, while the CsA group showed a significant increase in these levels. On the other hand, the rats co-treated with garlic (GAR+CsA) showed a substantial decrease in these levels compared to the CsA-treated rats. ALT and AST are known indicators of hepatic damage, and the ALP concentration is related to hepatocyte function [58]. The significant elevation of the serum levels of hepatic enzymes AST, ALT, and ALP in the CsA-treated groups in the present study may be due to the rapid release of these enzymes from the cytoplasm into the blood following plasma membrane rupture and parenchymal cellular damage [59,60]. High concentrations of serum transaminases are considered injurious to the liver [61]. Garlic supplementation caused a reduction in the ALT, AST, and ALP levels. These results agree with those of Ademiluyi et al. [62], who stated that garlic supplementation normalized the levels of AST and ALT after gentamycin toxicity. This may imply the ability of garlic to directly protect liver cells by stabilizing the cell membrane via prevention of hepatic glutathione depletion and lipid peroxidation [58]. The present study demonstrated that CsA decreases the concentration of albumin and total proteins (Table 1); total protein and albumin serum concentration reduced considerably in the CsA group rats, compared to the other groups. However, rats co-treated with garlic and CsA (GAR + CsA) exhibited substantial upregulation of these levels; this result concurs with Galán et al. [63], who found that CsA is hepatotoxic, and inhibits the synthesis of hepatic proteins. CsA administered orally to rats (15 mg/kg per day) for three weeks caused a marked decrease in the total serum levels of protein and albumin, accompanied by an increase in alkaline phosphatase. Enzymes (proteins) control different metabolic pathways and can damage other proteins, leading to changes in common metabolic pathways that cause tissue damage [61]. This process may also explain why the body weight decreased in rats treated with CsA. Our results demonstrate that garlic increases albumin and total protein levels compared to the CsA-treated group, and this result agrees with Mirunalini et al. [64], who stated that increased serum albumin was observed in raw garlic-treated patients. Stabilization of the serum proteins is a direct predictor of the improvement in liver cell function linked to garlic. Treatment with garlic treatment normalized the total proteins by stimulating protein synthesis, contributing to an improved hepatoprotective mechanism, and accelerated the regeneration process of liver cells [61].

The liver also plays a crucial role during lipid metabolism and various stages of lipid synthesis and transport. Therefore, an irregular lipid profile can be reasonably predicted in people with severe liver dysfunction [65]. Our results (Table 3) showed that the CsA group had significant increases in cholesterol, triglyceride, and LDL-c levels along with a substantial decrease in HDL-c levels. This confirms the findings of Hulzebos et al. [66], who reported that hypercholesterolemia and hypertriglyceridemia are common adverse side effects of CsA administration. In our study, garlic reduced the cholesterol, triglyceride, and LDL-c levels, and improved HDL-c levels in the co-treated group. The same result was observed by El-Kott et al. [67], who found that garlic inhibits cholesterol synthesis. Garlic has also been found to reduce cholesterol absorption in high-cholesterol diets and to cause significant decreases in the levels of LDL, cholesterol, and triglycerides in the liver. HDL cholesterol was substantially increased even after cholesterol absorption was inhibited. This may be the mechanism that leads to positive improvements in the plasma lipoprotein profile and lipid content [68]. The androgen steroid hormone, testosterone, is naturally formed in the body and is mainly secreted from the Leydig cells (>95%) and, to a lesser degree, the zone reticularis of the adrenal cortex and the ovaries in women. The principal male sex hormone is testosterone. It plays a critical role in producing reproductive and secondary sexual attributes, such as muscle growth and strength, bone mass, and body hair growth [69]. Testosterone is considered a marker of testis physiology. Our results, as presented in Table 4, showed that CsA decreased serum testosterone concentrations with non-significant variations in the PAL and GAR groups compared to the control group. This result was consistent with that of Rajfer et al. [70] which suggests the inhibitory effect of cyclosporine on testosterone production. These results may be due to the degenerating effect of CsA on the interstitial tissue of the testes (Figure 3G). The present study showed that garlic increased the serum testosterone level compared to the CsA group. This result is supported by our data in Table 5 that shows a significant increase in the quantity and quality of the sperm cells in the PAL and GAR groups compared to the CsA group, with a significant increase in the sperm cell abnormalities in CsA group compared to the other groups. These results agreed with those of Obidike et al. [71], who reported that garlic can improve spermatogenesis in male albino rats; however, a contradictory result was recorded by Hammami et al. [72], who claimed that garlic harmed the male reproductive function. Garlic contains zinc and can thus increase testosterone levels by inhibiting prolactin secretion, a testosterone inhibitor. Furthermore, zinc supplementation will increase the testosterone levels depending on baseline levels [73,74,75]. The histopathological findings show an improvement in the structure of the testes and confirm these results (Figure 3H). In the same context, the results of our semen analyses support our testosterone findings, in which garlic supplementation improved semen characteristics. This result was in line with that of Costello et al. [76], who reported that garlic has a constructive impact in restoring testicular tissue to its normal state.

Our results revealed that CsA decreased the serum levels of T3 and T4, as seen in Table 4. It was seen that the serum quantities of T3 and T4 displayed non-significant variations in the PAL and GAR groups compared to the control group. However, rats in the CsA group showed a significant decrease in these levels compared to their control group counterparts. Notably, rats in the GAR + CsA group showed significant upregulation in these hormone levels, although these were not identical to those of the control group. These results are consistent with those of Matsuda and Koyasu [77], who found that CsA inhibits the initial activation of T-lymphocytes. CsA seems to block the action of mitogens such as concanavalin A and the ionophore A23187, which triggers Ca^2+^ inflow, without alternating mechanisms. These observations indicate that CsA can block the Ca^2+^ dependent T-cell activation step. The Na^+^/H antiport is triggered by stimulation of the complex Ca^2+^-associated T3/T cell receptor [78]. Our results showed that garlic improved the serum levels of T3 and T4 in the CsA group (Table 3). These results agreed with those obtained by Tavakoli et al. [79]. This demonstrates the beneficial effect of garlic extract on thyroid function.

In addition to the benefits outlined previously, garlic contains vitamin A and allicin [80], where the former is considered a significant factor in animal development. In our study, the biochemical trial results obtained from the CsA group were also supported by the histopathological findings. CsA has adverse effects on the testes, such as damaged seminiferous tubules and arrested maturation of spermatogonia. The results also showed cells with irregular, dense nuclei in the lumina of some tubules. The CsA-treated group showed completely damaged seminiferous tubules with incomplete maturation of spermatogonia and vast interstitial space (Figure 3G). Previous studies have also demonstrated that CsA increases abnormal sperm rates, alters sperm chromatin structure and the composition of sperm head essential proteins, and manifests biochemical and histological alterations in the testes [81]. The group co-treated with garlic showed an improvement in the histopathological features of the testes (Figure 3H). These histopathological results agree with those of Ekeleme-Egedigwe et al. [82], who demonstrated that garlic oil administration restores normal testicular histology with normal spermatogenesis.

SOD is an essential antioxidant defense enzyme that catalyzes the dismutation of oxygen (O_2_) radical anions to hydrogen peroxide (H_2_O_2_) and oxygen radicals (O-) [21]. A substantial decrease in hepatic SOD_1_ expression was observed in CsA-treated rats compared to the control and other groups (Figure 2A). This decrease is due to the free radicals generated by CsA, which disturb the antioxidant status and ultimately lead to oxidative stress. These results agree with those of Tirkey et al. [83], who reported that CsA decreased the Glutathione and SOD levels. On the other hand, the group co-treated with garlic showed a considerable upsurge in the SOD_1_ gene in hepatic tissue of rats compared to the CsA group (Figure 2). Garlic has been reported to significantly decrease lipid peroxidation and increase endogenous antioxidants such as SOD [84]. As an antioxidant, garlic can antagonize the depletion of SOD by acting as a free radical scavenger that reduces the free radical load and stimulates SOD activity or expression, thus lessening oxidative injury to the tissues. By suppressing the formation of ROS and preserving antioxidant machinery, the innate capacity of liver cells to excite and sustain protection against oxidants by secreting more antioxidants is overwhelmed by the loss of oxidative stress. The genes synthesizing antioxidant enzymes, such as SOD, are simultaneously modulated in liver tissues [85]. The results shown in Figure 2A reveal that CsA upregulates fibrosis-related genes. This finding was consistent with several studies which reported that CsA invokes the pathogenesis of nephrotoxicity and may involve TGF-β1 and procollagen α1 [73,74]. The co-administration of garlic with CsA normalized the fibrosis-related genes. Our results agreed with those of D’Argenio et al. [86], who reported that garlic extract inhibited TGF-β1, alleviating liver fibrosis in rats. Additionally, garlic administration caused a significant upregulation of steroidogenesis genes, and the co-administration of garlic with CsA instigated a substantial increase in the STAR, CYP17A, and 3β-HSD genes. This result was consistent with that of Srinivas et al. [7] who demonstrated the harmful effect of CsA on rat fertility, focusing on the importance of garlic in male rat fertility [87], and showed that garlic might play a significant role in male Wistar rats’ reproductive processes. From our obtained result, we can reveal the possible mechanism of action of garlic supplementation that may be due to a reduction in ALT, AST, and ALP. It may imply the ability to protect liver cells directly. Garlic reduced the cholesterol, triglyceride, and LDL-c levels with improved HDL-c in co-treated and post-treated groups leads to positive improvements in plasma lipoprotein profile and lipid. A considerable upregulation in the SOD1 gene with normalized the fibrosis-related gene expression in rats’ hepatic tissue compared to the CsA treated group. Garlic supplementation increased the serum testosterone level and improved the semen characters with significant upregulation of steroidogenesis genes STAR, CYP17A 3B-HSD.

## 5. Conclusions

The administration of CsA may cause toxicity in the liver and testes of male rats. This study found that treatment with garlic may alleviate the toxic effects of CsA by normalizing the hepatic enzyme levels and strengthening the histological architecture of the liver. These improvements may be due to the antioxidant and anti-fibrogenesis properties of garlic, as seen by its modulation of SOD gene expression and the modulation of steroidogenesis genes, respectively.

## Figures and Tables

**Figure 1 animals-11-00064-f001:**
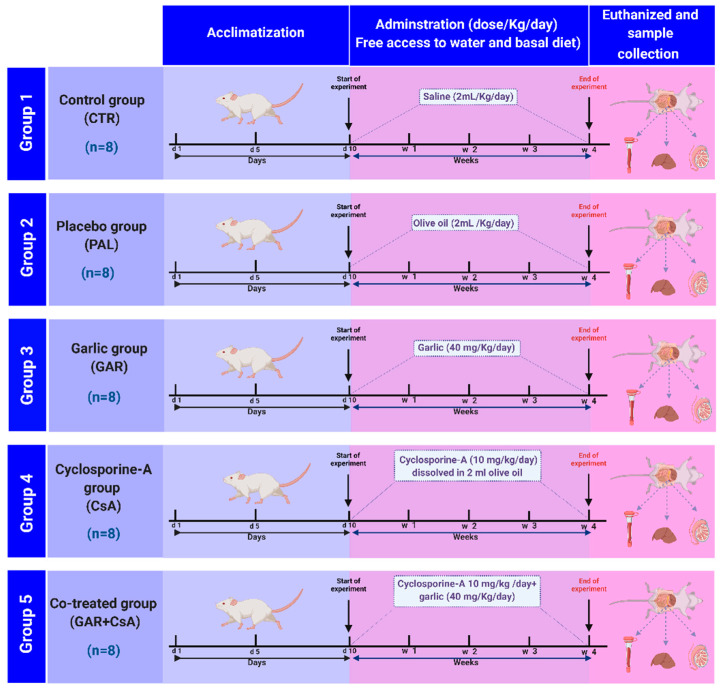
Experimental design.

**Figure 2 animals-11-00064-f002:**
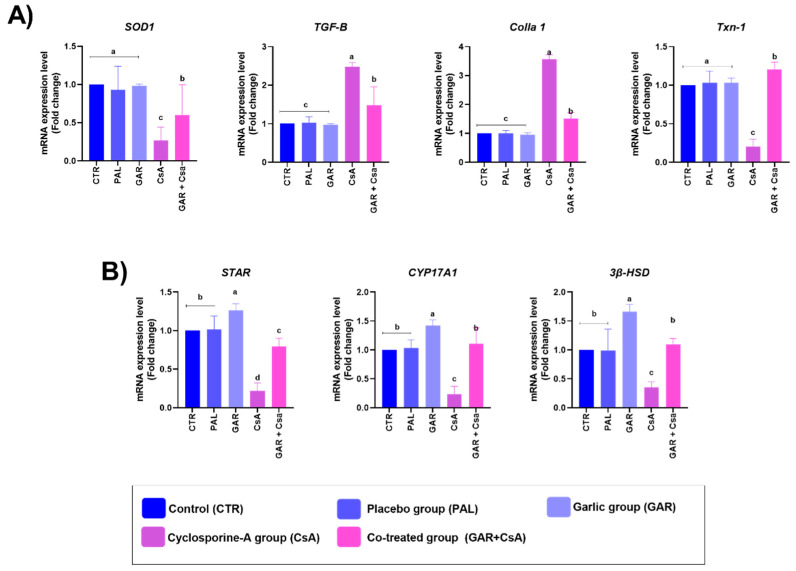
(**A**). Illustration of real-time quantitative PCR analysis of SOD_1_, TGF-β1, Colla1, and Txn1 gene expression in rat’s liver following Cyclosporine (CsA) and/or Garlic. (**B**). Illustration of real-time quantitative PCR analysis of STAR, CYP17A, and 3β-HSD gene expression in rat testes following administration of Cyclosporine (CsA) and/or Garlic. Data is expressed as mean ± SEM of 8 observations. Columns carrying different superscript letters are significantly different at *p* ≤ 0.05 compared to the control group for each gene expression.

**Figure 3 animals-11-00064-f003:**
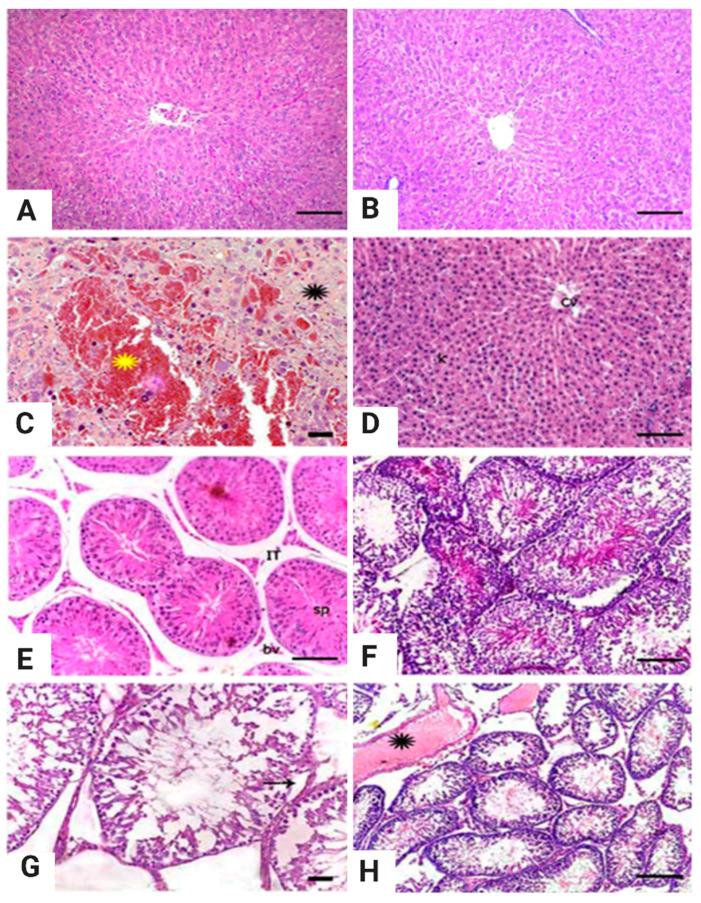
Representative photomicrograph for liver and testes tissues from Westar male albino rats treated with Cyclosporine (CsA) and/or Garlic for four weeks, H&E, scale bar = 200 µm (**A**,**B**,**D**), bar = 100 µm (**E**,**F**,**H**), bar = 50 µm (**C**,**G**). (**A**,**B**) Hepatic tissue of control (**A**) and garlic-treated rats (**B**) showing normal hepatic cords and central vein. (C) Liver tissues from a rat treated with CsA (10 mg/kg BW/day) showing area of hemorrhage (yellow asterisks) and necrotic hepatocytes (black asterisks). (**D**) Liver tissues of the co-treated group with CsA and garlic showing nearly normal hepatic tissue with the central vein. (**E**,**F**) Testicular tissues from control (**E**) and garlic-treated rats (**F**) showing normal histologic limits of seminiferous tubule lined by stratified germinal epithelium and contains luminal aggregations of sperms, the interstitium (IT) shows clusters of cells and blood vessels. (**G**) Testis from CsA treated rats showing damaged seminiferous tubules, maturation arrest up to the spermatogonia stage, Sertoli cell vacuolization (arrow), and wide intracellular space. (**H**) Testes from rat co-treated with CsA and garlic showing a mild improvement of the seminiferous tubules with some of them showing complete sperm maturation with congested blood vessels (asterisks).

**Table 1 animals-11-00064-t001:** RT-PCR primers for gene expression analysis.

Gene Symbol	Accession No.	Sequence (5′→3′)
SOD-1	NM_017050.1	F: TTTTGCTCTCCCAGGTTCCG
		R: CCCATGCTCGCCTTCAGTTA
TGF-1β	NM_021578.2	F: GGCCAGATCCTGTCCAAACT
		R: CGTGTTGCTCCACAGTTGAC
Txn-1	NM_053800.3	F: GGTAGTGGACTTCTCTGCCAC
		R: AGGTCGGCATGCATTTGACT
Col1a1	NM_053304.1	F: TTTCCCCCAACCCTGGAAAC
		R: CAGTGGGCAGAAAGGGACTT
STAR	NM_031558	F: CACACTTTGGGGAGATGCCT
		R: GAACTTCCAATGGCGTGCAG
CYP17A1	NM_012753.2	F: ACTGAGGGTATCGTGGATGC
		R: TCGAACTTCTCCCTGCACTT
3β-HSD	M38178	F: CCCATACAGCAAAAGGATGG
		R: GCCGCAAGTATCATGACAGA
β-actin	EF156276.1	F: TCTTCCAGCCTTCCTTCCTG
		R: CACACAGAGTACTTGCGCTC

SOD1, superoxide dismutase 1; TGF-1β, transforming growth factor-1β; Txn-1, Thioredoxin-1; Col1a1, collagen I-α1; STAR, steroidogenic acute regulatory protein; CYP17A1, cytochrome P450 17A1; 3β-HSD, 3-beta-hydroxysteroid dehydrogenase/delta-5-delta-4 isomerase type I.

**Table 2 animals-11-00064-t002:** Effect of treatment with Cyclosporine (CsA) and/or Garlic on serum levels of ALT, AST, ALP, albumin, and total protein of male Westar albino rats.

Treated Groups	ALT (IU/L)	AST (IU/L)	ALP (IU/L)	Albumin (g/dL)	Total Protein (g/dL)
CTR	65.77 ± 3.28 ^c^	151.52 ± 12.37 ^c^	120.50 ± 7.18 ^c^	4.02 ± 0.40 ^a^	7.05 ± 0.43 ^a^
PAL	67.40 ± 1.89 ^c^	152.58 ± 9.96 ^c^	121.48 ± 6.46 ^c^	3.97 ± 0.15 ^a^	6.85 ± 0.34 ^a^
GAR	68.50 ± 1.93 ^c^	154.42 ± 9.36 ^c^	121.18 ± 9.13 ^c^	4.11 ± 0.57 ^a^	6.94 ± 0.41 ^a^
CsA	98.52 ± 4.63 ^a^	205.00 ± 14.52 ^a^	182.28 ± 7.18 ^a^	3.24 ± 0.29 ^d^	4.59 ± 0.73 ^b^
GAR + CsA	73.70 ± 1.50 ^b^	160.62 ± 8.85 ^b^	163.28 ± 12.90 ^b^	3.38 ± 0.30 ^c^	5.04 ± 0.60 ^b^

Data are expressed as mean ± SEM of 8 observations. The statistical significance was set at *p* ≤ 0.05. Different superscript letters (^a^, ^b^, ^c^, and ^d^) indicate significant differences in the same column. ALT: Alanine aminotransferase; AST: Aspartate aminotransferase; ALP: Alkaline phosphatase; CTR: Control group; PAL: Placebo group; GAR: Garlic group; CsA: Cyclosporine-A group; GAR + CsA: Co-treated group with garlic and cyclosporine-A.

**Table 3 animals-11-00064-t003:** Effect of treatment with Cyclosporine (CsA) and/or Garlic on serum levels of total cholesterol, triglycerides, HDL-c, and LDL-c in male Westar albino rats.

Treated Groups	Total Cholesterol (mg/dL)	Triglycerides (mg/dL)	HDL-c (mg/dL)	LDL-c (mg/dL)
CTR	70.64 ± 6.98 ^c^	62.41 ± 4.206 ^c^	49.48 ± 5.37 ^a^	8.50 ± 2.95 ^d^
PAL	71.77 ± 6.77 ^c^	65.05 ± 2.69 ^c^	48.91 ± 4.02 ^a^	9.67 ± 3.23 ^d^
GAR	72.07 ± 3.38 ^c^	67.79 ± 2.95 ^bc^	46.51 ± 5.23 ^ab^	13.00 ± 6.41 ^cd^
CsA	97.51 ± 12.09 ^a^	88.28 ± 11.64 ^a^	35.06 ± 6.22 ^c^	44.86 ± 3.66 ^a^
GAR + CsA	85.79 ± 4.94 ^b^	79.73 ± 2.36 ^b^	41.77 ± 4.98 ^bc^	28.11 ± 5.31 ^b^

Data are expressed as mean ± SEM of 8 observations. The statistical significance was set at *p* ≤ 0.05. Different superscript letters (^a^, ^b^, ^c^, and ^d^) indicate significant differences in the same column. HDL-c: High-density lipoprotein cholesterol; LDL-c: Low-density lipoprotein cholesterol. CTR: Control group; PAL: Placebo group; GAR: Garlic group; CsA: Cyclosporine-A group; GAR + CsA: Co-treated group with garlic and cyclosporine-A.

**Table 4 animals-11-00064-t004:** Effect of treatment with Cyclosporine (CsA) and/or Garlic on serum levels of testosterone and concentrations of T3 and T4 in male Westar albino rats.

Treated Groups	Testosterone (ng/dL)	T3 (ng/dL)	T4 (µg/dL)
CTR	2.17 ± 0.09 ^a^	2.73 ± 0.03 ^a^	16.77 ± 0.819 ^a^
PAL	2.14 ± 0.07 ^a^	2.52 ± 0.07 ^ab^	16.35 ± 0.61 ^ab^
GAR	1.95 ± 0.06 ^ab^	2.58 ± 0.13 ^ab^	16.49 ± 0.50 ^ab^
CsA	0.72 ± 0.04 ^c^	1.46 ± 0.06 ^d^	12.53 ± 0.15 ^d^
GAR+CsA	1.41 ± 0.11 ^b^	1.80 ± 0.10 ^c^	14.56 ± 0.27 ^c^

Data are expressed as mean ± SEM of 8 observations. The statistical significance was set at *p* ≤ 0.05. Different superscript letters (^a^, ^b^, ^c^, and ^d^) indicate significant differences in the same column. T3: Triiodothyronine; T4: Thyroxine CTR: Control group; PAL: Placebo group; GAR: Garlic group; CsA: Cyclosporine-A group; GAR+CsA: Co-treated group with garlic and cyclosporine-A.

**Table 5 animals-11-00064-t005:** Effect of treatment with Cyclosporine (CsA) and/or Garlic on semen analysis.

Groups/Parameters	Control	PAL	GAR	CsA	GAR + CsA
Sperm cell count (× 10^6^/mL)	151.10 ± 2.51 ^a^	150.50 ± 3.52 ^a^	153.50 ± 4.50 ^a^	102.2 ± 4.49 ^c^	143.50 ± 3.51 ^ab^
Sperm motility%	92.00 ± 2.22 ^a^	91.00 ± 2.14 ^a^	92.00 ± 2.07 ^a^	72.10 ± 3.92 ^c^	85.00 ± 1.97 ^b^
Live spermatozoa%	94.10 ± 3.22 ^a^	92.10 ± 1.74 ^a^	93.00 ± 3.67 ^a^	76.12 ± 2.92 ^c^	87.12 ± 3.02 ^b^
Abnormality%	7.10 ± 0.41 ^c^	7.41 ± 0.88 ^c^	7.33 ± 0.33 ^c^	19.40 ± 0.71 ^a^	10.03 ± 0.98 ^b^

Data are expressed as mean ± SEM of 8 observations. The statistical significance was set at *p* ≤ 0.05. Different superscript letters (^a^, ^b^, ^c^, and ^d^) indicate significant differences in the same column.

## Data Availability

All data sets obtained and analyzed during the current study are available on fair request from the respective author.

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
