# Peer review of "Garlic Alleviates the Injurious Impact of Cyclosporine-A in Male Rats through Modulation of Fibrogenic and Steroidogenic Genes"

_animals, 2020, doi:10.3390/ani11010064_

Round 1

Reviewer 1 Report

Thank you for the opportunity to review the manuscript "Garlic alleviates the injurious impact of Cyclosporine-A on the male rat through modulation of he fibrogenic and steroidogenic genes". Studies of the natural products, as garlic, and their impact on health are important and less studied. 

However, the title of the manuscript sounds as we will find the explanation of  the mechanism of action, how garlic modulates fibrogenic and steroidogenic genes, but the manuscript only demonstrates the garlic impact on fibrogenic and steroidogenic genes. Even discussion section does not include considerations of possible mechanism of action.

Moreover, there is no explanation why testicular function was studies, while ciclosporin has the biggest impact on kidney and liver. In ciclosporin monography (after performed clinical trials) it is noted that adverse effects to reproductive system is rare and basically includes menstrual disturbances and gynecomastia. Toxicology studies on rates using 14 mg/kg/day (non-toxic ciclosporin level) showed no clinical adverse findings. Slight reduction in circulating lymphocytes after 3 weeks. Occasional erythrocytes in urinary sediment. Loose, divergent or overgrown incisors in several rats. Some lymphoid atrophy and slight adaptive changes in kidneys and livers of males. 

Ciclosporin doses 45-90 mg/kg/day (toxic ciclosporin level) showed lethal to 6/20 rats at mid and 18/20 rats at high dose levels due to hepatic and renal toxicity. After 6 weeks without drug, survivors’ BUN and SGPT returned to normal. Loosening of incisor teeth and hair loss. 

Why ciclosporin was chosen to induce toxicity for reproductive system, not the other medicaments better known for their toxicity to reproductive system?

If ciclosporin induce sperm toxicity on rats (novelty), what is the significance of this observation for human?

Other remarks:

-Simple summary, Abstract and Introduction

First time mentioned abbreviations should be explained. 

-Materials and Methods

Please explain, how were the doses calculated? 

-Results

"rats from saline (PAL) and Garlic (GAR)..." (157 line) sounds as rats are made from saline or garlic. Could be changed to rats receiving saline or garlic..

I would like to see letters a, b and c better explained in Tables descriptions.

"alterations" (183 line) - could be changed to variation/change.

I did not find figure 1A. I was not included in the manuscript I received.

"Treatment with saline..." (210). Saline or olive oil is not treatment. Could be changed to supplementation  / supplemented with...

Furthermore, there are a lot of grammatical errors, swapped letters, sentence punctuation errors. For example in 125, 159, 162, 188, 189, 261, 353 lines and etc.

Reviewer 2 Report

In this manuscript, the authors describes how " Garlic alleviates the injurious impact of Cyclosporine-A on the male rat through modulation of fibrogenic and steroidogenic genes". While this manuscript is generally well written, it would be helpful if the authors address the following issues:

1) In line 51, ROS should be defined instead of defining it in line 57.

2) In lines 113, 118, 119, 127, 227, 284, 291, 294, 334, 339, 357 etc, the manner in which the references are assigned can be confusing. This issue is widespread throughout the manuscript. It would be helpful if the authors revise them. Some helpful suggestion: x,y,x et al [24].

3) In line 197-198, the title of table 4 must be moved to the next page to avoid any potential distraction.

4) In line 336, an appropriate chemical formula of hydrogen peroxide should be used ie H2O2.

In short, the manuscript has the potential to benefit its targeted audience if the above issues raised are addressed.

Reviewer 3 Report

Fig. 1 missing.

That means the manuscript is incomplete.

To make decision, need all details.

Also, must explain all the results. Merely giving the Tables is NOt enough. Explain why and how for each table and tehir values.

Author Response

Manuscript ID: animals-1012610

Title: Garlic alleviates the injurious impact of Cyclosporine-A on the male rat through modulation of the fibrogenic and steroidogenic genes

Please find our revised manuscript entitled " Garlic alleviates the injurious impact of Cyclosporine-A on the male rat through modulation of the fibrogenic and steroidogenic genes''. We want to inform you that all our responses to the Reviewers' comments are included below in this covering letter.

             Thank you in anticipation.

                                                                                   Sincerely,

                              Mustafa Shukry, Associate professor, PhD, MD, BVs

                                Department of Physiology, Faculty of

                                Veterinary Medicine, Kafrelsheikh University,

                                 El-Geish Street, 33516 Kafrelsheikh, Egypt

                                 Telephone/Fax: +02-(0)47-323-1311

                                  E-mail address: [email protected]

Revision Note

Reply to the Reviewers

We wish to thank the Reviewers and Editor for these valuable comments on our submitted manuscript. The following are our responses to their comments.

To Reviewer # 3:

Comments

  1. English language and style are fine/minor spell check required.

Response: At first, we would like to thank the reviewer for this kind appreciation and great notion. Secondly, concerning the English language, a native English speaker revised the whole manuscript.

  1. Are the results clearly presented?/ Are the conclusions supported by the results?

Response : Thank you for your time and peer revision. We have modified the result as well as the conclusion.

  1. 1 missing. That means the manuscript is incomplete. To make decision, need all details.

Response: We would like to thank the reviewer for this kind appreciation and great notion., We already added it with the initial submission, and we insert it again within the manuscript as well as we added it in the response.

  1. Also, must explain all the results. Merely giving the Tables is NOt enough. Explain why and how for each table and their values.

Response: we want to thank you for your time and peer revision. We have modulated this and explain the result again(please, see the highlighted colored in the manuscript)

All comments in the pdf were corrected in the manuscript in equivalent position to it as mentioned in the pdf (please, see the highlighted colour throughout the whole manuscript).

  1. give details how/why it is antixoidant, and etc.give the chemical formula, chemical structure and the reactions/mechanisms of actions

Response

Thank you for your comments. We added the details in the manuscript(please, see the highlighted color line at 66-70)

  1. how old are these? and why that age is selected?

Response

approximately one and a half months of age relative to the weight recorded by Sengupta [1]. Moreover, Zemunik et al. [2]suggested that pubertal male rats are not fully mature and have not reached full reproductive capacity at 50–55 days of age. (please, see the highlighted color in the material section concerning the animals).

  1. why/how 2mL/kg? use kg, k lower case -in all places

Response

Thank you for your valuable comments.

The Garlic dose was chosen according to the previous studies [3,4]

- Garlic extract: 200 mg of garlic tablet were weighted by sensitive balance then homogenized with 10 ml saline to prepare a concentration of 40 mg/kg then take 2ml of this mixture to reach the desired dose of the cyclosporine(10mg/kg) then administered by oral gavage to each rat of the Cyclosporine group according to each weight  and the (Control group; CTR): Rats received saline (2mL/Kg/day)  Animals received saline (2mL /Kg/day) an equivalent amount of saline also according to the weight of each treated rat

The cyclosporine was according to [5,6]

The cyclosporine capsule (25mg) dissolved in 5 ml olive oil then take 2ml of this mixture to reach to the desired dose of the cyclosporine(10mg/kg) then administered by oral gavage to each rat of the Cyclosporine group according to each weight and the Group2 (Placebo one; PAL): Animals received olive oil (2mL /Kg/day) an equivalent amount of olive oil also according to the weight of each treated rat.

  1. "rats from saline (PAL) and Garlic (GAR)..." (157 line) sounds as rats are made from saline or garlic. Could be changed to rats receiving saline or garlic..

Response

Thank you for your comment.

Changed  (please, see the highlighted colour)

References

  1. Sengupta, P. The laboratory rat: relating its age with human's. International journal of preventive medicine 2013, 4, 624.
  2. Zemunik, T.; Peruzovic, M.; Capkun, V.; Zekan, L.; Tomic, S.; Milkovic, K. Reproductive ability of pubertal male and female rats. Brazilian journal of medical and biological research 2003, 36, 871-877.
  3. Mukherjee, D.; Banerjee, S. Learning and memory promoting effects of crude garlic extract. 2013.
  4. Sungnoon, R.; Kanlop, N.; Chattipakorn, S.C.; Tawan, R.; Chattipakorn, N. Effects of garlic on the induction of ventricular fibrillation. Nutrition 2008, 24, 711-716.
  5. Luo, X.; Yang, T.; Yang, C.; Zhou, J.; Liu, Y.; Huang, Y.; Shi, S. Effects of multiple oral dosing of cyclosporine on the pharmacokinetics of quercetin in rats. Int J Clin Exp Med 2016, 9, 5880-5890.
  6. Wassef, R.; Cohen, Z.; Langer, B. Pharmacokinetic profiles of cyclosporine in rats. Influence of route of administration and dosage. Transplantation 1985, 40, 489-493.

Round 2

Reviewer 1 Report

I suggest to make some additional corrections to the manuscript regarding the authors' provided answers:

1) Additional explanation why CsA was chosen for this study can be added.

2) Detailed information regarding the doses of CsA, garlic used in study should be added to the method section. To make data reproduction simply for future studies. 

Other answers were provided adequately. Thank you. 

Reviewer 3 Report

Good work to study garlic's use.

Please include the mechanism of action of garlic.

Author Response

Reply to the Reviewers

We wish to thank the Reviewers and Editor for these valuable comments on our submitted manuscript. The following are our responses to their comments.

Reviewer # 3:

Comments

Comments and Suggestions for Authors

Good work to study garlic's use.

Please include the mechanism of action of garlic.

Response: we want to thank you for your time and peer revision.

We added it to the manuscript (please, see line 459-466 )